# Prenatal Low-Protein Diet Affects Mitochondrial Structure and Function in the Skeletal Muscle of Adult Female Offspring

**DOI:** 10.3390/nu14061158

**Published:** 2022-03-09

**Authors:** Vipin A. Vidyadharan, Ancizar Betancourt, Craig Smith, Chandrasekhar Yallampalli, Chellakkan S. Blesson

**Affiliations:** 1Basic Sciences Perinatology Research Laboratories, Department of Obstetrics and Gynecology, Baylor College of Medicine, Houston, TX 77030, USA; vidyadha@bcm.edu (V.A.V.); ancizarb@bcm.edu (A.B.); cyallamp@bcm.edu (C.Y.); 2Agilent Technologies Inc., Santa Clara, CA 95051, USA; craig.smith@agilent.com; 3Reproductive Endocrinology and Infertility Division, Department of Obstetrics and Gynecology, Baylor College of Medicine, Houston, TX 77030, USA; 4Family Fertility Center, Texas Children’s Hospital, Houston, TX 77030, USA

**Keywords:** developmental programming, glucose intolerance, insulin resistance, mitochondria, low-protein diet

## Abstract

Gestational low-protein (LP) diet leads to glucose intolerance and insulin resistance in adult offspring. We had earlier demonstrated that LP programming affects glucose disposal in females. Mitochondrial health is crucial for normal glucose metabolism in skeletal muscle. In this study, we sought to analyze mitochondrial structure, function, and associated genes in skeletal muscles to explore the molecular mechanism of insulin resistance LP-programmed female offspring. On day four of pregnancy, rats were assigned to a control diet containing 20% protein or an isocaloric 6% protein-containing diet. Standard laboratory diet was given to the dams after delivery until the end of weaning and to pups after weaning. Gestational LP diet led to changes in mitochondrial ultrastructure in the gastrocnemius muscles, including a nine-fold increase in the presence of giant mitochondria along with unevenly formed cristae. Further, functional analysis showed that LP programming caused impaired mitochondrial functions. Although the mitochondrial copy number did not show significant changes, key genes involved in mitochondrial structure and function such as Fis1, Opa1, Mfn2, Nrf1, Nrf2, Pgc1b, Cox4b, Esrra, and Vdac were dysregulated. Our study shows that prenatal LP programming induced disruption in mitochondrial ultrastructure and function in the skeletal muscle of female offspring.

## 1. Introduction

The developmental origin of the health and disease hypothesis postulates the importance of the fetal environment during development [1,2]. Maternal diet is a vital factor that determines the trajectory of growth and early development of the offspring [2,3]. Many epidemiological and experimental studies indicate that maternal low-protein (LP) diet is linked to an increased risk of metabolic diseases in adulthood [4,5,6]. As prenatal LP diet-exposed offspring are found to be insulin resistant, the prevalence of type 2 diabetes (T2D) is much higher in them when compared with normal offspring. Maternal LP diet causes low levels of amino acids in fetal circulation, thereby depriving the developing fetus of key amino acids, leading to low birth weight and impaired skeletal muscle development [7,8].

Skeletal muscles play a crucial role in energy homeostasis by burning glucose and fatty acids. This process is aided by the presence of mitochondria in the skeletal muscles. The metabolic ability of skeletal muscles is unique, as they can adapt quickly to various functional demands, and this ability is largely due to the presence of a healthy mitochondrial population [9,10]. Thus, the quality and quantity of mitochondria in skeletal muscle are critical for glucose homeostasis [11]. Skeletal muscle is the primary site of peripheral insulin resistance, as several steps of insulin-stimulated glucose uptake and utilization take place in this tissue [12]. Peripheral insulin resistance is the hallmark of impaired glucose homeostasis and is a predictor of T2D [13]. Various studies have indicated that insulin resistance is often associated with alterations in skeletal muscle mitochondrial health caused by either a reduced number or dysfunctional mitochondria [14,15].

Prenatal nutrition is crucial for the normal development of skeletal muscles, and its health is determined by the quality and quantity of muscle fibers formed during fetal development [16,17]. Maternal undernutrition affects not only the number of muscle fibers but also the functional attributes of the skeletal muscles and stem cell activity [18,19,20]. Previous studies using maternal LP diet-based animal models have reported long-lasting changes in mitochondrial function [21,22,23]. Further, prenatal LP diet followed by postnatal high-fat diet resulted in reduced skeletal muscle mitochondrial oxidation [21]. Another study shows that fetal and early postnatal protein restriction caused decreased mtDNA in the skeletal muscles and downregulation of mitochondrially encoded genes [22]. In addition, a microarray analysis on skeletal muscles of newborn mice exposed to LP during gestation revealed a significant reduction in the expression of mitochondrial genes involved in oxidative phosphorylation [23]. As skeletal muscle functions as one of the main sites for peripheral glucose disposal, the mitochondrial dysfunction in these tissues often leads to insulin resistance and glucose intolerance [24].

In our earlier studies, we had extensively characterized a gestational LP-induced lean T2D rat model. We showed that pups born to LP dams are smaller at birth but show rapid catch-up growth [25,26,27]. They developed progressively worsening glucose intolerance and insulin resistance with advancing age without any increase in body weight, body mass index, or fat content when compared with controls [25,28]. We also showed that gestational LP diet caused T2D in adult offspring with different signaling mechanisms in males and females [25,26,27,29,30]. Our studies in females showed that impaired glucose homeostasis is caused due to a defect in the insulin signaling cascade, which regulates glycogen synthesis [26]. Further, we also showed that LP programming affects hepatic glucose production via both gluconeogenesis and glycogenolysis in a sex-dependent manner with greater dysregulation in females when compared to males [29]. The present study aims to investigate the role of mitochondrial health in the skeletal muscle in LP-programmed T2D female offspring. We hypothesized that a gestational LP diet impairs mitochondrial ultrastructure and expression of genes associated with mitochondrial biogenesis and dynamics along with compromised mitochondrial function in female adult offspring.

## 2. Materials and Methods

### 2.1. Animals

All the animal procedures were approved by the Institutional Animal Care and Use Committee of the Baylor College of Medicine, Houston, Texas. Outbred Wistar rats were obtained from Envigo bioproducts Inc. Madison, WI. Virgin females weighing ~250 g and males weighing ~350 g were purchased and housed in a temperature-controlled room (~23 °C) with a 10:14 h light/dark cycle and were given unlimited access to food and water. Female rats were mated with males of proven fertility by housing two females with one male. Females were checked for the presence of sperm in vaginal smear, and the presence of sperm was marked as day one of pregnancy. Custom-formulated feed pellets were obtained from Harlan Teklad (Madison, WI) in consultation with their in-house nutritionist. The control diet (Diet # TD.91352) had 20% protein (casein), and the LP diet (Diet # TD.90016) had 6% protein (casein). Diets were made isocaloric (3.8 kcal/g) by adjusting the carbohydrate quantity. Control diet had 20.3% (21.5% kcal) protein, 61.6% (65.3% kcal) carbohydrates, and 5.5% (13.1% kcal) fat, and LP diet had 6.1% (6.5% kcal) protein, 75.6% (80.4% kcal) carbohydrates, and 5.5% (13.3% kcal) fat. Further, the diets were matched for calcium (0.7%) and phosphorus (0.54%). On day four of pregnancy, rats were randomly assigned to a control diet containing 20% protein (*n* = 10) or an isocaloric 6% protein-containing diet (*n* = 10) until delivery. Standard laboratory rat chow (Teklad Global (Diet # 2019) containing 19% Protein Extruded Rodent Diet, Teklad Diets, Madison, WI, USA) was given to dams after delivery until the end of weaning, and pups were given the standard laboratory rat chow after weaning. Two-day-old pups were sexed, and pups with extreme weights were culled to normalize the litter size to eight pups (four males and four females when possible) per mother in both groups. Dams were euthanized after weaning. All experiments were performed using 4-month-old rats except for the Seahorse XF Cell Mito Stress Test, where the rats were 6 months old. Rats were euthanized during diestrus, and tissues were harvested and processed for various studies.

### 2.2. Transmission Electron Microscopy (TEM)

TEM was carried out as described previously [31]. Briefly, gastrocnemius muscles were dissected and were placed in fixative (2.5% glutaraldehyde in 0.1M cacodylate buffer to a pH of 7.4 at 4 °C) and trimmed into small pieces. The pieces were placed in the fixative and stored at 4 °C until processing. The muscle pieces were then washed in buffer for 15 min thrice followed by post fixing in 1% OsO_4_ in 0.1M cacodylate for 45 min at 4 °C. The samples were washed three times in distilled water, followed by washes in ascending series of alcohol. The muscle samples were then embedded and cured in resin. Fifty nanometer sections were then cut from these blocks using an ultra-microtome. The thin sections were mounted on a copper grid and stained with heavy metals for ultra-structural analysis. For image analysis, 10 randomly chosen fields were photographed at 3000× for each animal, and the mitochondria were manually counted for five animals for each group. Selected areas were imaged at 10,000× to observe the mitochondrial ultrastructure.

### 2.3. Mitochondrial DNA Copy Number

Total DNA was isolated from gastrocnemius muscle using QIAmp DNA kit (Qiagen, Germany). The genomic DNA was stored at −80 °C for subsequent use. Mitochondrial DNA copy number was assessed as previously described [31,32]. The qPCR was performed with 1:100 of diluted DNA templates for mitochondrially encoded cytochrome c oxidase 1, 2, and 3 and compared with genomically encoded reference gene, tubulin. The total volume of reaction conducted was 10 μL, and the reaction mix was prepared fresh every time. The PCR conditions used were 3 min at 95 °C for initial denaturing, followed by 15 s at 95 °C, 30 s at 60 °C for annealing, and 15 s at 72 °C for extension for 40 cycles, followed by a melt curve analysis. All samples were run in triplicates. Details of primers are provided in Table 1.

### 2.4. Quantitative Real-Time qPCR

Expression of key genes involved in mitochondrial function was quantified using qPCR. Total RNA was isolated from gastrocnemius muscle by using TRIzol reagent (Life Technologies, Carlsbad, CA, USA). Total RNA was further refined with the RNeasy Cleanup Kit (Qiagen, Valencia, CA, USA). All the RNA samples were quantified using the ND-1000 model NanoDrop Spectrophotometer (Thermo Fisher Scientific, Newark, DE, USA). Total RNA (2 µg) was reverse transcribed using a modified Maloney murine leukemia virus-derived RT (New England Biolabs Inc., Ipswich, MA, USA) and random hexamer primers (Life Technologies) as reported earlier [26]. After dilution, cDNA was amplified by real-time PCR using SYBR Green (Bio-Rad, Hercules, CA, USA) in a CFX96 model real-time thermal cycler (Bio-Rad). Specific pairs of primers were designed and purchased (IDT, Coralville, IA, USA). Details of primers are provided in Table 1. All reactions were performed in triplicates, and cyclophilin A was used as the reference gene. Results were calculated using 2^–ΔΔ^CT method and expressed as fold changes of expression of genes of interest.

### 2.5. Mitochondrial Oxygen Consumption

The FDB muscles were isolated, and Cell Mito Stress Test was performed using an XF 96 Extracellular Flux Analyzer (Seahorse Bioscience) as described earlier [33]. Briefly, harvested FDB muscle pieces (~20 μg) from each rat were incubated in dissociating media at 5% CO_2_ and 37 °C for 2 h. Dissociation media were prepared by dissolving collagenase (4 mg/mL; Roche, Mannheim, Germany) in DMEM (Invitrogen, Carlsbad, CA, USA) containing 2% charcoal-stripped FBS along with 0.1% gentamycin. After dissociation, single myofibers were separated from each FDB muscle bundle. After removing the undigested fragments, myofibers were transferred to a 35 mm sterile dish with 1 mL of culture media (DMEM with 2% charcoal-stripped FBS with gentamycin). After thoroughly dispersing the muscle fibers, 50 μL aliquots of the fibers were taken and seeded into the Aligent Seahorse XFe96 microplate pre-coated with extracellular matrix (ECM, Sigma-Aldrich, St. Louis, MO, USA) to facilitate attachment of the muscle fibers. The fibers covered 60–70% of the well bottom. Samples from each animal were seeded in triplicates. The microplates were placed in a 5% CO_2_ incubator at 37 °C overnight before analysis. Mitochondrial stress tests were performed by following manufacturer’s protocol (Agilent Technologies). XF DMEM medium was supplemented with 10 mM glucose solution, 2 mM glutamine solution, and 1 mM pyruvate for the assay. Inhibitors were used at the following concentrations: 2.5 µM oligomycin, 0.5 mM FCCP, and 0.5 µM antimycin A + 0.5 µM rotenone. Data analyses were conducted using Wave software and XF Report Generators (Agilent Technologies). OCR measurements were normalized to protein content.

### 2.6. Western Blot

Western blots for mitochondrial proteins in gastrocnemius muscle were performed as described earlier [34]. Briefly, 30 μg of protein extract was resolved on 4–15% precast gradient polyacrylamide gels (Mini-PROTEAN TGX Precast Gels; Bio-Rad, Hercules, CA, USA). Resolved proteins were transferred to a polyvinylidene fluoride membrane (Millipore, Billerica, MA, USA). Primary antibodies were incubated overnight at 4 °C after blocking the membranes in 5% bovine serum albumin or 5% nonfat dried milk in Tris-buffered saline containing 0.1% Tween 20 for 1 h at room temperature. Details of primary antibodies and their dilutions are as follows: Gapdh (Cat #97166, 1:1000), Vdac1 (Cat #4661, 1:1000), Opa1 (Cat #80471, 1:1000), Nrf1 (Cat #46743, 1:1000), Sirt1 (Cat #9475, 1:1000), Esrra (Cat #13826, 1:1000), and Cox-IV (Cat #4850, 1:5000) were obtained from Cell Signaling Technology (Danvers, MA, USA); Fis1 (Cat #sc-376447, 1:1000), Mfn1 (Cat #sc-166644, 1:1000), and Mfn2 (Cat #sc-515647, 1:1000) were obtained from Santa Cruz Biotechnology (Dallas, TX, USA); and Total OXPHOS Rodent Antibody Cocktail (Cat #ab110413) and Pgc1b (Cat # ab 176328) were obtained from Abcam Cambridge (MA, USA). After primary antibody incubations, membranes were washed and incubated for 60 min at room temperature with horseradish peroxidase-conjugated secondary antibodies (Proteintech Inc., Rosemont, IL, USA). Membranes were washed and incubated in ECL Western blotting detection reagents (Pierce Biotechnology, Waltham, MA USA) for detection and imaged using the Odyssey Fc imaging system (LI-COR). Densitometry analyses were performed using Image Studio software from LI-COR.

### 2.7. Statistical Analysis

Statistical analyses were performed using GraphPad Prism software. Data are presented as the mean ± SEM. Comparisons between the two groups were performed using unpaired Student *t*-tests. Differences were considered to be statistically significant when *p* < 0.05.

## 3. Results

### 3.1. Controls and LP Rats Had Similar Weights at 4 Months and Feed Intake

LP-programmed and control female offspring had similar body weights at 4 months of age (LP: 329.5 ± 8 vs. Control: 348.8 ± 12 g, *p* > 0.05). Further, there were no differences between the weights of gastrocnemius muscles between the two groups (LP: 1.8 ± 0.05 vs. Control: 1.9 ± 0.05 g, *p* > 0.05). Feed intake normalized to their body weight at 4 months of age did not show differences between the LP-programmed females and controls (LP: 17.5 ± 1.5 vs. Control: 19.0 ± 1.0 g/kg body weight, *p* > 0.05).

### 3.2. Maternal LP Diet Alters the Mitochondrial Morphology and Ultrastructure

TEM images showed that LP programming caused changes to mitochondrial morphology and ultrastructure in gastrocnemius muscles when compared with controls (Figure 1A,B). Although there were no differences in the total number of mitochondria per field (Figure 1C), LP-programmed muscles showed an eight-fold increase in the number of giant mitochondria (Figure 1D), clearly showing anomalies in the fission-fusion process of mitochondria. Further, control animals had well-defined cristae with an electron-dense matrix that was distributed uniformly throughout the mitochondria, whereas the LP-programmed animals had an electron-dense matrix with sparsely distributed and unevenly formed cristae, which could lead to compromised mitochondrial function.

### 3.3. LP Programming Did Not Alter mtDNA Copy Number in Gastrocnemius Muscles

Although image analysis from TEM did not show any changes in mitochondrial numbers, we wanted to further confirm by assessing the mtDNA copy number. The mtDNA copy number was assessed by qPCR for genes encoded in mitochondria (mtCo1, mtCo2, mtCo3) and compared with a somatic reference gene (β-actin) in LP-programmed gastrocnemius muscles versus controls. No difference noted in mtDNA copy number of LP-programmed skeletal muscle when compared with controls for mtCo1 (LP: 251 ± 18 vs. Control: 279 ± 23), for mtCo2 (LP: 1331 ± 51 vs. Control: 1349 ± 195), and for mtCo3 (LP 580 ± 38 vs. Control 586 ± 71) (Figure 2).

### 3.4. LP Programming Reduced Oxygen Consumption in Skeletal Muscles

The Cell Mito Stress Test was conducted to examine the impact of LP programming on the change in oxygen consumption rate (OCR) and mitochondrial function (Figure 3). Overall results suggest that skeletal muscles (FDB) from LP-programmed females showed a decrease in OCR (Figure 3A). Further analysis using various inhibitors and the algorithm of Seahorse Bioscience Inc. for the Mito Stress Test showed that there was no difference in the OCR during basal respiration (Figure 3B; 15.5 ± 9.8 in controls vs. 14.1 ± 9.0 pmol/min/mg protein in LP). However, there were significant decreases in the OCR during maximum respiration (Figure 3C; 66.6 ± 9.6 in controls vs. 31.7 ± 6.4 pmol/min/mg protein in LP, *p* < 0.05), non-mitochondrial respiration (Figure 3D; 56.7 ± 5.8 in controls vs. 39.1 ± 3.6 pmol/min/mg protein in LP, *p* < 0.05), and spare respiratory capacity (Figure 3E; 45.5 ± 5.6 in controls vs. 21.7 ± 4.9 pmol/min/mg protein in LP, *p* < 0.05) clearly showing compromised mitochondrial function.

### 3.5. LP Programming Reduced the Expression of Mitochondrial Complex I

We further wanted to investigate whether the molecular mechanisms leading to mitochondrial oxidative phosphorylation were impaired. To determine whether LP programming altered the ETC protein levels, we measured the OXPHOS-associated protein complexes in the muscle. We found that the protein levels of Complex 1 (Ndufb8) were reduced around 1.5-fold (*p* < 0.05) in the LP muscles compared with the control (Figure 4A,B). The protein levels of other complexes (Complex II, Complex III, Complex IV, and Complex V) did not show any significant changes (Figure 4C–F).

### 3.6. LP Programming Dysregulated Mitochondrial Dynamics and Biogenesis Genes

Given the physiological and structural attributes of mitochondria in the LP muscles, we further investigated the role of mitochondrial dynamics and biogenesis in compromised mitochondrial function. First, we determined the mRNA expression of key genes involved in mitochondrial dynamics and biogenesis in the skeletal muscle using qPCR (Figure 5). Analyses were performed for mitochondrial fusion genes *Opa1*, *Mfn1*, and *Mfn2*, and fission genes *Drp1* and *Fis1*. Among the mitochondrial fusion genes tested, the expression levels of *Opa1* and *Mfn2* in the LP skeletal muscles were significantly (*p* < 0.05) reduced than in the controls (Figure 5A,C). However, the *Mfn1* expression level did not show any change between the LP and the control (Figure 5B). The mitochondrial fission gene *Fis1* expression was significantly (*p* < 0.05) increased in the LP skeletal muscles compared with the control (Figure 5E), but *Drp1* expression remained the same between the groups (Figure 5D).

The expression of key genes involved in mitochondrial biogenesis that were investigated included *Nrf1*, *Nrf2*, *Parl1*, *Pgc1a*, *Pgc1b*, *Sirt1*, *Esrra*, *Cox1*, *CoxIVa*, *Cox IVb*, and *Vdac1*. Among these genes, *Pgc1b*, *Esrra*, and *Nrf1* were significantly (*p* < 0.05) downregulated in the LP-programmed rats (Figure 5F–H). Similarly, the expression of *Sirt1* and *Vdac1* were also downregulated (*p* < 0.01) in LP-programmed rats when compared with controls (Figure 5J,K). On the other hand, *Nrf2* and *Cox IVb* expressions were significantly (*p* < 0.05) elevated in the LP skeletal muscles than in the controls (Figure 5I,L). Other investigated genes did not show any differences (data not shown).

We further investigated the protein expression pattern of these genes (Figure 6A). Their expression pattern reflected a similar pattern to that of mRNA. The inner mitochondrial fusion gene Opa1 (*p* < 0.01) and outer mitochondrial membrane fusion gene Mfn2 (*p* < 0.05) were downregulated in LP-programmed females (Figure 6B,D) with no changes in Mfn1 (Figure 6C). However, the mitochondrial fission gene Fis1 protein level was significantly (*p* < 0.01) increased in the LP skeletal muscle when compared with controls (Figure 6E). Further, the mitochondrial biogenesis-linked genes such as Pgc1b, Nrf1, Esrra, and Vdac1 levels were significantly (*p* < 0.05) reduced in the LP skeletal muscles compared with the control (Figure 6F–I). However, CoxIV and Sirt1 protein levels were not different between the groups (Figure 6J,K).

## 4. Discussion

Skeletal muscle is a major site of glucose disposal where glucose is broken down to produce energy. We had earlier shown that LP-programmed T2D rats had compromised insulin signaling leading to peripheral insulin resistance [25,26]. It is well known that mitochondria play a vital role in energy production, and mitochondrial dysfunction has been observed in insulin target tissues during insulin-resistant states [35]. Abnormal mitochondria are often connected to metabolic syndrome [36]. Various reports show that mitochondrial structure, number, and oxidative capacity are diminished in skeletal muscles of patients with insulin resistance [37,38]. Further, insulin resistance and T2D are often associated with the dysregulation of genes responsible for oxidative metabolism and dysfunctional mitochondrial electron transport chain in skeletal muscles [11,39,40,41,42]. Skeletal muscle fibers are the center of glucose and fatty acid utilization; therefore, defective mitochondria affect the peripheral glucose disposal capacity [43,44].

Gestational LP diet is known to dysregulate mitochondrial function in skeletal muscles leading to insulin resistance [23,45,46,47]. Further, mitochondrial genes involved in oxidative phosphorylation were downregulated in skeletal muscles, indicating LP programming could contribute to mitochondrial dysfunction [23]. In the present study, we report for the first time that LP programming during gestation causes significant changes in mitochondrial morphology, ultrastructure, and function without changes to the mitochondrial copy number in the skeletal muscles of adult female offspring in a lean T2D rat model. Structural abnormalities in mitochondria have been reported in protein-restricted animal models [48,49,50]. The morphological changes to mitochondria in the skeletal muscle exposed to in utero LP diet may indicate impaired mitochondrial dynamics [51].

Mitochondrial morphology and ultrastructure often mirror the functional characteristics of the mitochondrial network in a cell. We needed to utilize FDB muscles to perform functional analysis due to technical difficulties with the large muscle fibers of gastrocnemius muscles. Although the quantity of mitochondria is different between the skeletal muscle types, the mitochondrial oxidative capacity and mitochondrial efficiency was found to be similar [52,53,54,55]. These data suggest most mitochondrial qualitative properties are conserved across different fiber types. Hence, FDB is a reasonable alternative; however, the data should be interpreted with caution. Functional analysis using Mito Stress Test showed that LP skeletal muscles displayed inferior mitochondrial function than the controls. Interestingly, mitochondrial respiration in the presence of FCCP (maximal respiration) was lower in LP animals compared with control animals. The depletion of maximal respiratory capacity indicates the incompetence of skeletal muscle to meet any additional ATP demand, which is often observed under the settings of electron transport chain inhibition [56]. Interestingly, the depletion in maximal respiratory capacity has been reported in the skeletal muscle mitochondria of type 2 diabetic individuals [57,58]. As the number of mitochondria was similar in both the groups, impaired electron transport chain components might be the cause of diminished maximal respiratory capacity. Spare respiratory capacity is the ability to increase the metabolic rate from baseline for accommodating transiently elevated energy demands [59,60]. Therefore, the reduction in spare respiratory capacity shown by the LP skeletal muscle further indicates the intrinsic defect with the LP mitochondria and the reduced ability to adapt during increased energy demand.

To further understand the role of mitochondrial complexes in the mitochondrial structure and function, we determined the levels of different electron transport chain (ETC) complexes in the skeletal muscle. A downregulation of genes responsible for oxidative phosphorylation and dysfunctional mitochondrial electron transport chain was observed in earlier studies in other models and tissues [11,23,39]. In our study, we noticed maternal LP diet programming caused a reduction in the Complex I protein, Ndufb8. Mitochondrial Complex I is composed of 44 different subunits and one of the largest membrane-bound protein complexes in the ETC [61]. Complex I deficiency is found to be one of the common biochemical defects in metabolic diseases and mitochondrial disease in children [62]. In addition, Complex I is one of the key components of mitochondrial super complex, which is essential for normal oxidative phosphorylation [63]. Therefore, the reduction in Complex I proteins might limit the super complex’s ability to perform optimally. Our result is consistent with a previous study that reported impaired mitochondrial super complex assembly and function in the muscles fibers of type 2 diabetes patients [64].

We also show that the expression of key genes involved in mitochondrial dynamics and biogenesis is dysregulated in the skeletal muscles. Mitochondrial dynamics is a tightly choreographed action of mitochondrial fusion (Opa1 and Mfn) and fission (Drp1 and Fis1) proteins [51]. Earlier studies have clearly shown that abnormal mitochondrial dynamics affect normal glucose disposal [36,43,44]. Opa1 is obligatory at the inner mitochondrial membrane (IMM) for tethering and fusion of membranes, and it is also critical for cristae remodeling [65]. Moreover, Opa1 levels decreased with the progression of insulin resistance [66]. The two isoforms of mitofusin (Mfn1 and Mfn2) cooperate and aid in mitochondrial outer membrane fusion. Mfn2 expression is greater in muscle compared with other tissues, and it is crucial for the sustenance and operation of the mitochondrial network [67,68]. Previous studies have noticed that Mfn2 is a crucial player in maintaining glucose homeostasis [41]. Mfn2 regulates insulin signaling and sensitivity in muscle and is often downregulated in the skeletal muscle of T2D patients [69,70]. A positive correlation between Mfn2 expression and improved insulin sensitivity via improved Glut4 translocation has also been reported [71]. Further, Mfn2 deficiency is frequently coupled with mitochondrial dysfunction and impaired mitochondrial dynamics [41]. Fis1 is a mitochondrial receptor protein that helps to recruit Drp1 for mitochondrial fission. Independent of Drp1 protein association, Fis1 can also promote fission through functioning as a negative regulator of fusion machinery, Opa1 and Mfn2 [72]. Our study shows a significant reduction in Opa1 and Mfn2 expression in the skeletal muscle, and this could be responsible for the abnormal mitochondrial structure and could potentially contribute to insulin resistance. Our data show that Fis1 expression is increased in the LP samples compared with the controls, indicating a possible imbalance in the mitochondrial fission mechanism. However, another key gene involved in the fission process Drp1 showed no increase. The lack of Drp1 response might justify the presence of giant mitochondria in the LP samples, considering the critical role of Drp1 in mitochondrial fission [73]. Although inadequate, it is likely that increased Fis1 was a compensatory attempt to reduce Opa1 and Mfn2 and to promote mitochondrial fission [72]. Further, maternal LP diet altered the expression of several genes associated with mitochondrial biogenesis [23]. The transcriptional coactivator Pgc1b is a positive regulator of mitochondrial function and biogenesis in skeletal muscle [74]. Pgc1b is essential for adequate expression of genes associated with ETC and oxidative phosphorylation (OXPHOS), which regulate basal energy homeostasis [75]. Overexpression of Pgc1b in skeletal muscle was shown to improve diet-induced insulin resistance in rats [76]. As a co-transcription activator, Pgc1a/b induces mitochondrial biogenesis through the activation of a variety of transcription factors such as Esrra, Nrf1, and Nrf2 [77]. Esrra is an orphan nuclear receptor that targets several gene networks connected to mitochondrial biogenesis and glucose homeostasis [77]. Nrf1 and Nrf2 are critical factors to the series of events steps leading to enhance the transcription of important mitochondrial enzymes. In addition, they are found to interact with the mitochondrial transcription factor that facilitates transcription and mtDNA replication [78]. Furthermore, Nrf2 expression has been shown to reduce blood glucose levels and skeletal muscle glycogen content [79]. We had previously reported that the glycogen synthesis in skeletal muscle of LP female rats were inhibited [26]. It is likely that the downregulation of Pgc1 might lead to reduced Esrra, Nrf1, and Nrf2 expression, which could further impact the mitochondrial complex protein levels and establish a vicious metabolic cycle with the onset of insulin resistance. Reduced Pgc1 protein level is observed in skeletal muscles of T2D patients due to hypermethylation of the promoter [40,80]. Our data show Pgc1b levels were downregulated in the LP skeletal muscle, which indicates that mitochondrial biogenesis is impaired. The mismatch between the in utero and postnatal environments can introduce histones modifications in the Pgc1 promoter of skeletal muscle [81]. Although further studies are warranted to confirm the promotor methylation status of Pgc1 in our model, previous studies indicate that epigenetic changes may play a major role in reduced Pgc1 expression and associated mitochondria dysfunction [81,82]. Further, the adaptive response of skeletal muscle to energy demand is often synchronized through increased mitochondrial biogenesis, which is, in turn, regulated by complex networks of genes, including Pgc1, Esrra, Nrf1, and Nrf2 [40,41]. The dampened responses of these genes found in our study indicate that the mitochondrial adaptive response arm in the LP skeletal muscle is crippled by in utero LP programming. On the other hand, Pgc1a levels, most studied homolog of Pgc1b, were not different between the groups, suggesting that Pgc1b plays a major role in muscle mitochondrial biogenesis. Comparable results were reported in a previous study in which they found Pgc1b is more potent in stimulating mitochondrial biogenesis and respiration compared with Pgc1a [83]. Moreover, under basal (non-exercise stimulated) settings, the Pgc1b in skeletal muscle is found to be expressed 28 times higher than that of Pgc1a [41,84]. Hence, the unstimulated condition in the present study explains the diminished role of Pgc1a.

## 5. Conclusions

In summary, our results indicate that prenatal LP programming induced disruption in mitochondrial morphology and function along with dysregulation of genes involved in mitochondrial biogenesis and dynamics in the skeletal muscle of female offspring. In addition, LP programming compromised cellular bioenergetics by reducing maximal respiration, spare respiratory capacity, and mitochondrial Complex I level. We show that LP diet-induced mitochondrial dysfunction could be the uterine environment influenced modification of genes, which predisposes the offspring to mitochondrial dysfunction in adulthood (Figure 7). Taken together, our study shows maternal protein restriction triggered mitochondrial dysfunction in the skeletal muscle of female offspring, possibly leading to peripheral insulin resistance. Therefore, efforts to improve in utero nutrition and mitochondrial health could have a preventive capability in insulin resistance associated with maternal protein restriction.

## Figures and Tables

**Figure 1 nutrients-14-01158-f001:**
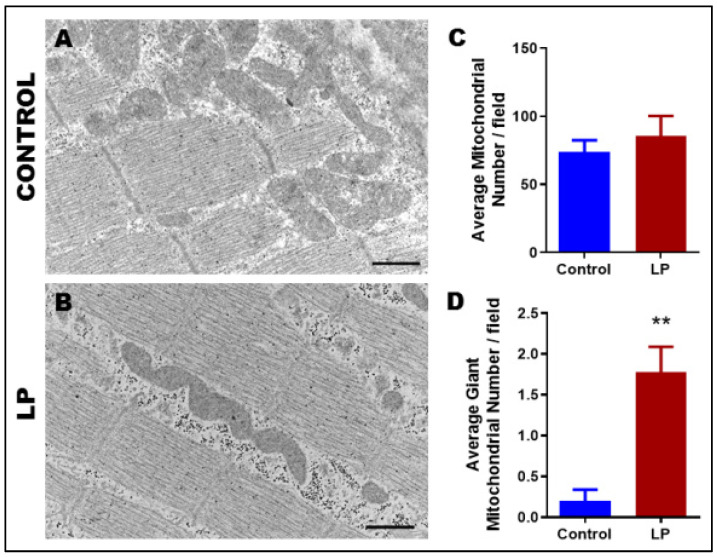
Transmission electron microscopic images depicting alteration of mitochondrial structure in control vs. LP (Low protein programmed lean diabetic rats). (**A**) Ultrastructure of mitochondria in the gastrocnemius muscle of control and (**B**) low protein programmed lean diabetic rats; (**C**) Average number of mitochondria/ field in control and LP rats; (**D**) Average number of giant mitochondria/field in control and LP rats (** *p* < 0.01); *n* = 5. Scale bar represent 800 nm.

**Figure 2 nutrients-14-01158-f002:**
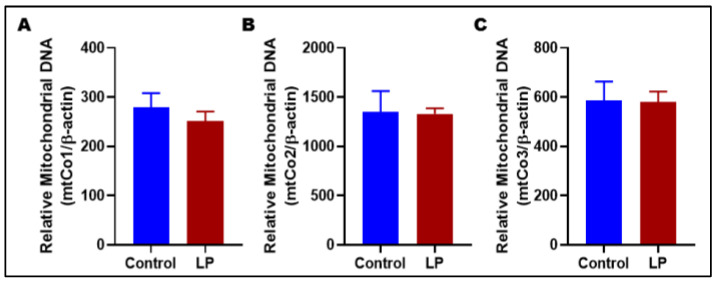
Mitochondrial copy number levels in control vs LP. Quantification of mitochondrial DNA copy number was conducted using qRT-PCR: (**A**) depicts mitochondrial Complex I (mtCo1) levels; (**B**) shows mtCo2 levels; (**C**) illustrates mtCo3 levels when normalized to beta-actin. Data represent mean ± SEM; *n* = 8.

**Figure 3 nutrients-14-01158-f003:**
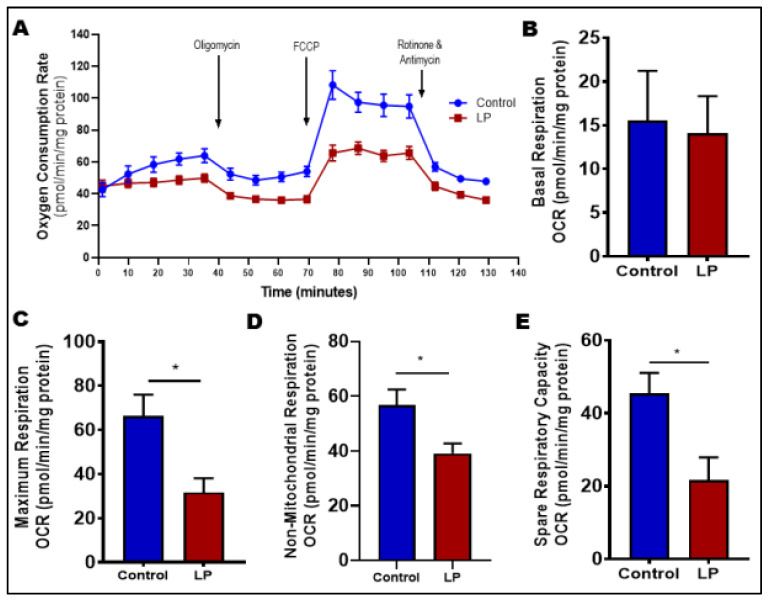
Mitochondrial oxygen consumption rate in control vs. LP skeletal muscle. Mitochondrial respiratory parameters in the skeletal muscle were measured using mitochondrial stress test. Basal respiration was measured before the addition of inhibitors. The arrows indicate the exact time at which different inhibitor compounds were injected into the wells: (**A**) representative image of normalized mitochondrial oxygen consumption rates; control (blue) vs. LP (red); (**B**) basal respiration; (**C**) maximal respiration; (**D**) non-mitochondrial respiration; (**E**) spare respiratory capacity. Data represent mean ± SEM (* *p* < 0.05); *n* = 8. FCCP: Carbonyl cyanide 4-(trifluoromethoxy)phenylhydrazone, OCR: Oxygen consumption rate.

**Figure 4 nutrients-14-01158-f004:**
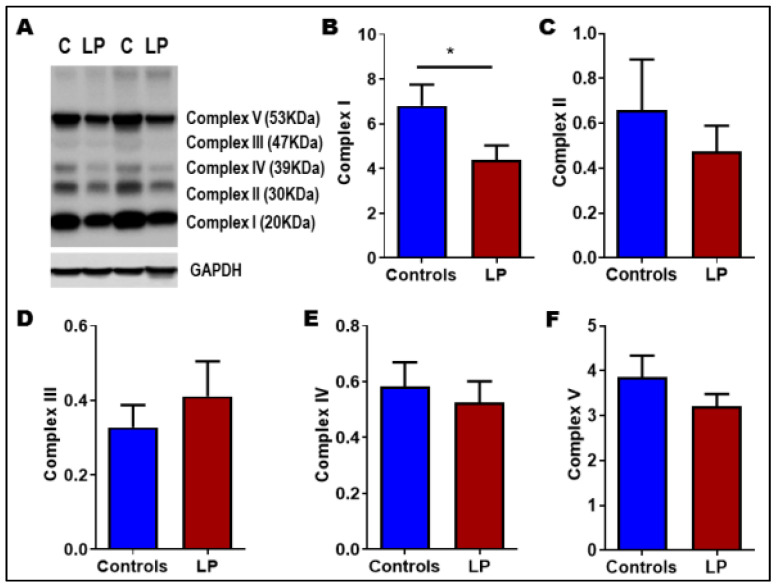
Mitochondrial respiratory complexes levels in control vs. LP skeletal muscle: (**A**) representative image of different mitochondrial complex protein content; (**B**–**F**) fold-change of protein levels for the different subunits of mitochondrial complexes: (**B**) Complex I (CI, Ndufb8); (**C**) Complex II (CII, Sdhb); (**D**) Complex III (CIII, Uqccrc2); (**E**) Complex IV (CIV, mtCo); Complex V (CV, ATP5a). GAPDH (Glyceraldehyde-3-phosphate dehydrogenase) was used as a loading control for Western blots, and its values were used for normalization. Data represent mean ± SEM (* *p* < 0.05,); *n* = 7.

**Figure 5 nutrients-14-01158-f005:**
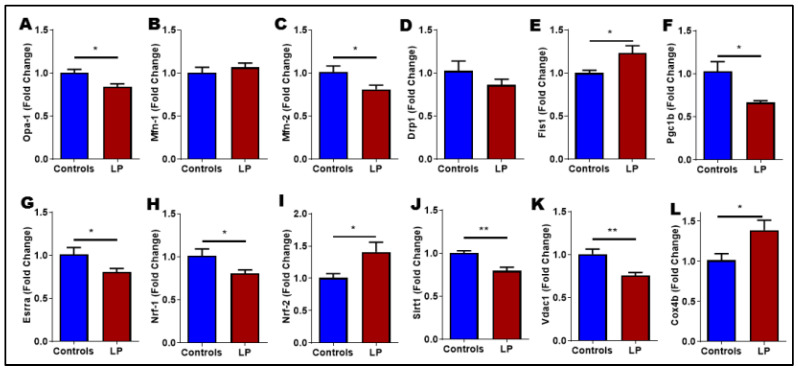
Expression of genes involved in mitochondrial dynamics and biogenesis in the skeletal muscle of control and LP offspring. qPCR analysis was conducted to determine expression of mitochondrial dynamic genes: (**A**) Opa1; (**B**) Mfn1; (**C**) Mfn2; (**D**) Drp1; (**E**) Fis1; and biogenesis genes: (**F**) Pgc1b; (**G**) Erra; (**H**) Nrf1; (**I**) Nrf2; (**J**) Sirt1; (**K**) Vdac1; (**L**) Cox 4b. The mRNA expression was normalized relative to Cyclophilin A expression. Data represent mean ± SEM (* *p* < 0.05, ** *p* < 0.01); *n* = 8.

**Figure 6 nutrients-14-01158-f006:**
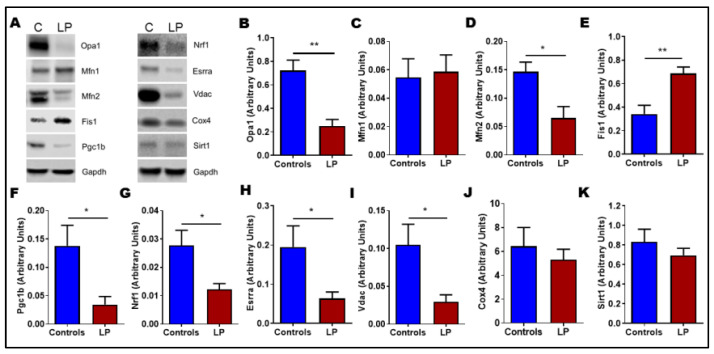
Change in mitochondrial dynamics and biogenesis protein content in the skeletal muscle of control and LP offspring: (**A**) representative Western blots of mitochondrial dynamics and biogenesis genes tested in controls (C) and low-protein (LP) group; (**B**–**K**) depicts relative expression of different genes: (**B**) Opa1; (**C**) Mfn1; (**D**) Mfn2; (**E**) Fis1; (**F**) Pgc1b; (**G**) Nrf1; (**H**) Erra; (**I**) Vdac1; (**J**) Cox 4; (**K**) Sirt1. The blot density was calculated by densitometry scanning and normalized to GAPDH. Data represent mean ± SEM (* *p* < 0.05, ** *p* < 0.01); *n* = 8.

**Figure 7 nutrients-14-01158-f007:**
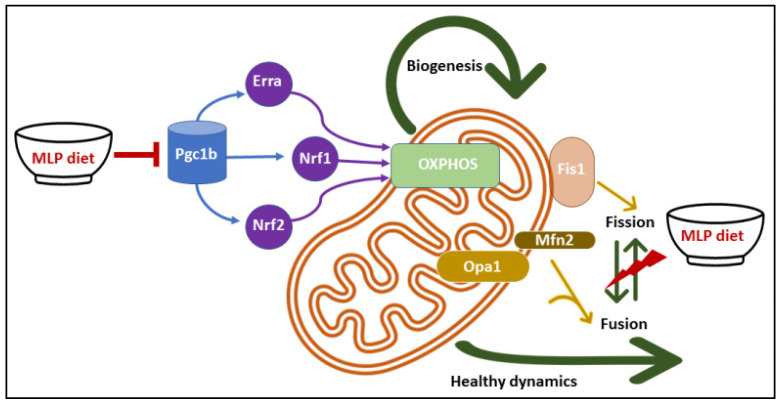
Proposed molecular mechanism of maternal low-protein diet (MLP) induced mitochondrial dysfunction in skeletal muscles.

**Table 1 nutrients-14-01158-t001:** Oligonucleotide primers used for qPCR.

Gene	Primer (F = Forward; R = Reverse)
*mtCox1*	F: 5′- ATCGCAATTCCTACAGGCGT-3′ R: 5′-TGTTAGGCCCCCTACTGTGA-3′
*mtCox2*	F: 5′-CAAGACGCCACATCACCTATC-3′ R: 5′-TTGGGCGTCTATTGTGCTTG-3′
*mtCox3*	F: 5′-GGAACATACCAAGGCCACCA-3′ R: 5′-TCGTGGGTAGGAACTAGGCT-3′
*Esrra*	F: 5′- AAAGTCCTGGCCCATTTCTATG-3′ R: 5′-CCCTTGCCTCAGTCCATCAT-3′
*CoxIVa*	F: 5′-CAAGGGCACCAATAGGTGGA-3′ R: 5′-GATGGGGCCATACACCTAGC-3′
*CoxIVb*	F: 5′-CGTCTTCAGCTTGCAACTATGT-3′ R: 5′-ACATAGGGGGTCATCCTCCG-3′
*Cyc A*	F: 5′-TATCTGCACTGCCAAGACTGAGTG-3′ R: 5′-CTTCTTGCTGGTCTTGCCATTCC-3′
*Fis1*	F: 5′-GTGCCTGGTTCGAAGCAAATA-3′ R: 5′-CATATTCCCGCTGCTCCTCTT-3′
*Mfn1*	F: 5′-ATCTTCGGCCAGTTACTGGAGTT-3′ R: 5′-AGATCATCCTCGGTTGCTATCC-3′
*Mfn2*	F: 5′-CCTTGAAGACACCCACAGGAATA-3′ R: 5′-CGCTGATTCCCCTGACCTT-3′
*Nrf1*	F: 5′-CTCTGCATCTCACCCTCCAAAC-3′ R: 5′-TCTTCCAGGATCATGCTCTTGTAC-3′
*Nrf2*	F: 5′-CATTTGTAGATGACCATGAGTCGC-3′ R: 5′-GAGCTATCGAGTGACTGAGCC-3′
*Opa1*	F: 5′-AAAAGCCCTTCCCAGTTCAGA-3′ R: 5′-TACCCGCAGTGAAGAAATCCTT-3′
*Pgc1a*	F: 5′-CTACAATGAATGCAGCGGTCTT-3′ R: 5′-TGCTCCATGAATTCTCGGTCTT-3′
*Pgc1b*	F: 5′-TCGGTGAAGGTCGTGTGGTATAC-3′ R: 5′-GCACTCGACTATCTCACCAAACA-3′
*Sirt1*	F: 5′-CTGTTTCCTGTGGGATACCTGACT-3′ R: 5′-ATCGAACATGGCTTGAGGATCT-3′
*Vdac1*	F: 5′-GTCACCGCCTCCGAGACCAT-3′ R: 5′-CCAATCCATTCTCGGACTTCGT-3′
*Tuba1a*	F: 5′-ATGGTCTTGTCGCTTGGCAT-3′ R: 5′-CCCCTTTCCACAGCGTGAGT-3′

## Data Availability

The data presented in this study are available on request from the corresponding author.

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
