# Peer review of "Prenatal Low-Protein Diet Affects Mitochondrial Structure and Function in the Skeletal Muscle of Adult Female Offspring"

_nutrients, 2022, doi:10.3390/nu14061158_

Round 1

Reviewer 1 Report

Vidyadharan et al. demonstrate key changes in mitochondrial structure and functional due to a low protein diet during gestation in mice.  Overall, the study is rigorously executed and clearly communicated in this manuscript. Some minor comments -

1.How translational is a 6% protein containing diet during gestation in the context of humans?

2.The authors should elaborate on possible mechanisms by which mitochondrial size and function is being dysregulated.

3.As the authors correctly point out - several studies demonstrate a decrease in mitochondrial copy number. Please speculate on why authors did not see such a change in this particular study. 

4.Please clarify on whether a low protein diet starting on day 4 of pregnancy is optimal. Several similar studies begin a deficient diet several weeks prior to pregnancy to ensure deficiency, especially due to the influence of epigenetic regulation. 

Author Response

Vidyadharan et al. demonstrate key changes in mitochondrial structure and functional due to a low protein diet during gestation in mice.  Overall, the study is rigorously executed and clearly communicated in this manuscript. Some minor comments -

We thank the reviewer for the positive comments and insightful questions. We have answered them accordingly.

  1. How translational is a 6% protein containing diet during gestation in the context of humans?

We appreciate the reviewer comments. Our diet of 6% is at the lower end of most of the animal models used in programming related studies. We give the modified diet for a short duration only during pregnancy. Other models employ long times before pregnancy and during lactation. Further, during pregnancy it is recommended to have 20-25% of one’s calorie from protein, however many of the pregnant women in developing countries do not have access to such high protein diet. We also employed this to mimic low protein during famine. We have extensively used this diet regimen and our model is characterized well and have published extensively showing all the programming effects leading to type 2 diabetes.  

  1. The authors should elaborate on possible mechanisms by which mitochondrial size and function is being dysregulated.

We thank the reviewer for this question. We think the mitochondrial morphology changes observed in the study may confer a short-term survival advantage for the cells in low protein environment during early development. However, this may incline towards disease susceptibility in different nutritional environment in the adult hood. We have mentioned this in conclusion section.  Further studies are warranted for identify the specific mechanisms leading to the mitochondrial abnormal size and function as observed in our study.  

  1. As the authors correctly point out - several studies demonstrate a decrease in mitochondrial copy number. Please speculate on why authors did not see such a change in this particular study. 

We thank the reviewer for this question. Although the total mitochondrial content is important in cells, equally important is how efficient or good these mitochondria are, specifically in their respiratory function. Mitochondrial copy number is typically used as a surrogate measure for the number of mitochondria present. In our case although the fission and fusion of mitochondria is affected, it did not translate into change in copy number. Instead, we observed compromised mitochondrial structure and function. This could be a tissue specific or time specific issue. Perhaps more importantly, it could also be that in LP programmed offspring, mitophagy is inefficient in destroying abnormal mitochondria.

  1. Please clarify on whether a low protein diet starting on day 4 of pregnancy is optimal. Several similar studies begin a deficient diet several weeks prior to pregnancy to ensure deficiency, especially due to the influence of epigenetic regulation. 

We thank the reviewer for this very important question. We designed the model in a way that low protein diet affects nutrition of the developing fetus. Since the implantation in rat takes place on day 5, we started treatment from day 4. We wanted to do the intervention only from the time of implantation to delivery, but not before and after to avoid other confounding factors. We have characterized this rat model over the years and indeed found this treatment regimen to be very effective and consistent.

Reviewer 2 Report

Vidyadharan et. al. in their work have studied the effect of maternal low protein diet during pregnancy on the offsprings. Through their previous works they got clue to study effects on mitochondria. The mitochondria count and DNA content are similar in the control and low protein diet group, however the morphology and mitochondrial dynamics varied. Subsequently, they quantified the transcripts and protein concerned with mitochondrial fission and fusion.

The study is well presented and is logically coherent. However, the link between low protein diet and mitochondrial genes expression is not presented and is left for future work.

In the introduction in line 42, it will be helpful if an estimate is given on number of mitochondria rather than using adjectives.

There are some typos on the usage of FDB ( somewhere it is FBD and at one place it is FDG).

How did they classify mitochondria as normal or giant? Was it only through visual inspection? 

I am guessing mitochondrial fission will be regulated along with mitochondrial DNA replication. It is surprising to see the DNA content is same but the mitochondria reached to the unsuccessful fission stage. 

What do the authors think why certain protein levels didn't mirror the change in their corresponding transcript level (for example COX 4b)?

Author Response

Vidyadharan et. al. in their work have studied the effect of maternal low protein diet during pregnancy on the offsprings. Through their previous works they got clue to study effects on mitochondria. The mitochondria count and DNA content are similar in the control and low protein diet group, however the morphology and mitochondrial dynamics varied. Subsequently, they quantified the transcripts and protein concerned with mitochondrial fission and fusion. The study is well presented and is logically coherent. However, the link between low protein diet and mitochondrial genes expression is not presented and is left for future work.

We thank the reviewer for all the positive observations. As observed by the reviewer, we do not yet know the link between low protein diet and mitochondrial abnormalities as observed here as of now. Currently we are exploring possible mechanisms but this requires more future investigations.

  1. In the introduction in line 42, it will be helpful if an estimate is given on number of mitochondria rather than using adjectives.

We thank the reviewer for raising this issue; it is hard to give an accurate estimate as it varies spatio-temporally and across tissue types. It’s content is relatively more in high energy utilizing tissues. We have now revised this sentence to “This process is aided by the presence of mitochondria in the skeletal muscles.”

  1. There are some typos on the usage of FDB (somewhere it is FBD and at one place it is FDG).

We thank the reviewer for this observation. We have now corrected the typos.

  1. How did they classify mitochondria as normal or giant? Was it only through visual inspection? 

Yes, the abnormal size and structure of the mitochondria was determined by visual inspection of TEM images by experienced investigators. 

  1. I am guessing mitochondrial fission will be regulated along with mitochondrial DNA replication. It is surprising to see the DNA content is same but the mitochondria reached to the unsuccessful fission stage. 

We agree with the reviewer on the importance of mitochondria DNA replication and mitochondrial dynamics. Mitochondrial DNA copy number is only a surrogate measurement for mitochondrial copy number. This is not an ideal measurement as each mitochondria can contain anywhere between 2-10 mtDNA copies. Hence mitochondrial copy number need not necessarily reflect actual mitochondrial number. It is likely that in low protein programmed animals, since the mitochondria are not dividing normally, there is probably no mtDNA replication and hence the overall number remains the same (As evident by counting mitochondrial numbers after TEM).

  1. What do the authors think why certain protein levels didn't mirror the change in their corresponding transcript level (for example COX 4b)?

We agree with the reviewer’s observations that protein levels did not mirror with the mRNA levels. This is a common occurrence and could be attributed to various aspects such as mRNA stability, protein stability, post-transcriptional regulation of gene via miRNAs, translational blockers etc.